# Application of Non-Destructive Testing Techniques (NDTT) to Characterize Nanocarriers Used for Drug Delivery: A Mini Review

**Rahul Islam Barbhuiya** [1], **Saipriya Ramalingam** [1], **Harsimran Kaur Kalra** [2], **Abdallah Elsayed** [1], **Winny Routray** [3], **Manickavasagan Annamalai** [1] **and Ashutosh Singh** [1,*]

[1]  School of Engineering, University of Guelph, Guelph, ON N1G 2W1, Canada; rbarbhui@uoguelph.ca (R.I.B.); sramalin@uoguelph.ca (S.R.); aelsay01@uoguelph.ca (A.E.); mannamal@uoguelph.ca (M.A.)

[2]  Department of Biomedical Sciences, Ontario Veterinary College, University of Guelph, Guelph, ON N1G 2W1, Canada; hkaur15@uoguelph.ca

[3]  Department of Food Process Engineering, National Institute of Technology, Rourkela 769008, India; routrayw@nitrkl.ac.in

*  Correspondence: asingh47@uoguelph.ca; Tel.: +1-519-824-4120

**Abstract:** The synthesis of tailored and highly engineered multifunctional pharmaceutical nanocarriers is an emerging field of study in drug delivery applications. They have a high surface-area-to-volume ratio, aiding the targeted drug's biodistribution and pharmacokinetic properties. Therefore, the characterization of nanocarriers is critical for understanding their physicochemical properties, which significantly impact their molecular and systemic functioning. To achieve specific goals, particle size, surface characteristics, and drug release properties of nanocarriers must be managed. This mini review provides an overview of the applications of non-destructive testing techniques (NDTT) to reveal the characteristics of nanocarriers, considering their surface charge, porosity, size, morphology, and crystalline organization. The compositional and microstructural characterization of nanocarriers through NDTT, such as dynamic light scattering, X-ray diffraction, confocal laser scanning microscopy, ultraviolet-visible spectroscopy, scanning electron microscopy, atomic force microscopy, and nuclear magnetic resonance spectroscopy, have been comprehensively reviewed. Furthermore, NDTT is only used to characterize physicochemical parameters related to the physiological performance of nanocarriers but does not account for nanocarrier toxicity. Hence, it is highly recommended that in the future, NDTT be developed to assess the toxicity of nanocarriers. In addition, by developing more advanced, effective, and precise techniques, such as machine vision techniques using artificial intelligence, the future of using NDTT for nanocarrier characterization will improve the evaluation of internal quality parameters.

**Keywords:** characterization; drug delivery; non-destructive testing; NDTT; nanocarriers; nanomaterials

## 1. Introduction

In recent years, nanotechnology has gained considerable application in the field of medicine and engineering. It studies and manages matter at the nanoscale level with diameters ranging from 1 to 100 nanometers [1]. It deals with different nanoparticles derived from polymers, metals, non-metals, or combinations [1]. These nanoparticles are employed in various applications, from electronics to drug delivery [2].

The nanoparticles used for drug delivery, known as nanocarriers, offer several benefits, such as the ability to deliver hydrophobic and hydrophilic drug molecules, precision regarding the target, and a high level of stability [3]. Nanocarriers are colloidal chemicals (drugs, pesticides, fertilizers, and plant growth promoters) that can modify their bioactivation and properties [4]. They have a high surface-to-volume ratio, which can aid in the biodistribution and pharmacokinetics of the targeted compounds [4]. Nanoparticles,

nanospheres, nanoemulsions, nanocapsules, and nano-sized vesicular carriers, such as niosomes and liposomes, can all be considered as nanocarriers [5]. Nanospheres are matrix particles in which pharmaceuticals are uniformly distributed, whereas nanocapsules contain a distinct polymeric membrane surrounding the chemicals of interest, such as drugs, proteins, and additives [4,6]. Moreover, nanocarriers can be categorized into two types, as shown in Figure 1, based on the constituents used for their manufacture: polymer-based or dendritic and lipid-based nanocarriers. Solid lipid nanoparticles, nanoemulsions, nanostructured lipid carriers, and liposomes are examples of lipid-based nanocarriers. On the other hand, nano-colloids, polymer micelles, dendritic branches, polymer vesicles, and polymer nanoparticles are polymer and dendritic branch-based nanocarriers [4].

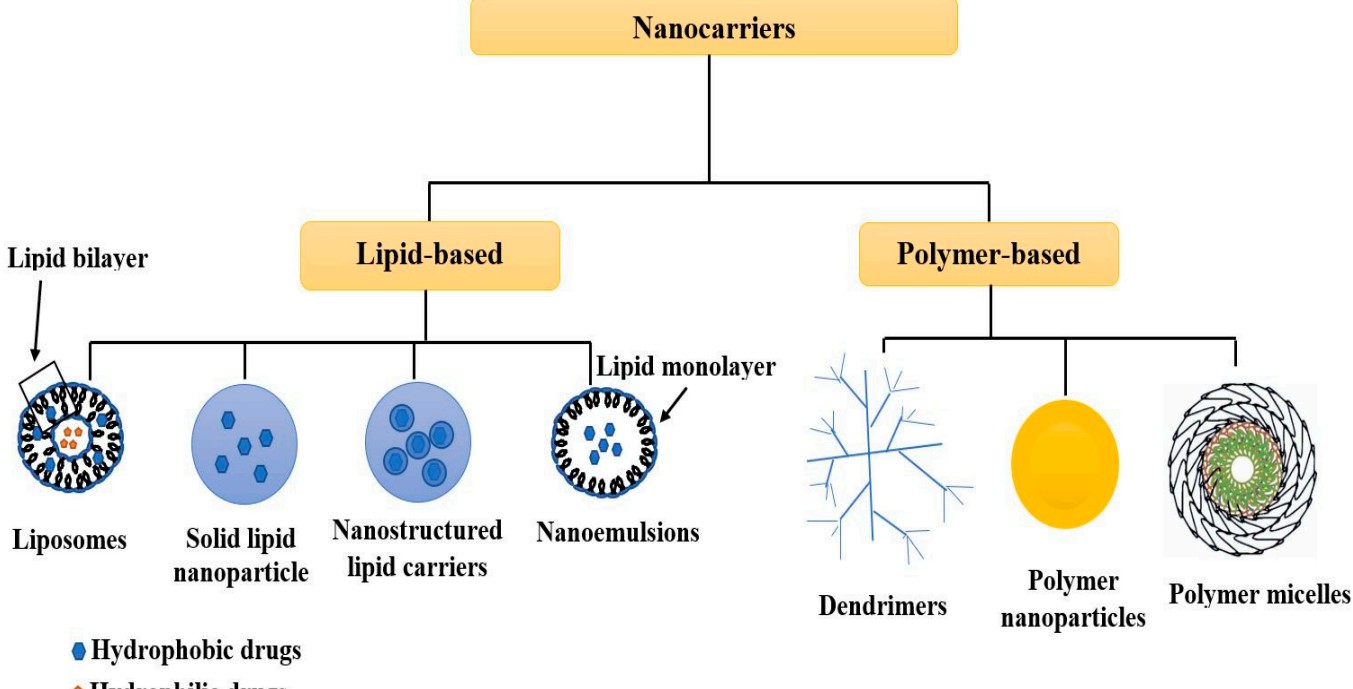

**Figure 1.** Schematic representation of different lipid-based and polymer-based nanocarriers.

Furthermore, it is essential to comprehend how our biological system will respond to nanocarriers. As particle size shrinks, surface area relative to volume increases, allowing for increased particle interaction with its surroundings [7]. Consequently, nanocarrier size and surface area strongly influence the fate of nanoparticles within the body. When it comes to nanocarrier retention, distribution, absorption, and elimination, the lymphatic system and the blood capillaries play a significant role [8]. The lymph nodes detect any foreign substance moving through the body as the fluids are reabsorbed into the blood. If the particles are larger, the immune system identifies them as foreign, and the macrophages consume them before expelling them from the body [9]. This is a challenge for nanosized drug delivery systems; nevertheless, the clearance of the medication is also influenced by the drug's particle size and surface features, which are equally as essential as the formulation of the drug. Hence, nanocarriers must regulate proper drug release properties, surface properties, and particle size to achieve specified goals. So, researchers need to use advanced characterization methodologies in the early stages of product development to correlate the product's effects with their biological implications and to forecast the outcomes of therapeutic interventions [10].

When it comes to pharmaceutical development, traditional characterizing methods, such as reverse phase high-performance liquid chromatography (RP-HPLC), typically necessitate destructive extraction procedures on small, nominally random product samples to document the product's quality [11,12]. Such techniques are time consuming and fre-

quently ineffective at ensuring zero-defect product quality because they fail to identify the effect of crucial process parameters on product quality features [13]. Therefore, researchers proceeded forward and began characterization using non-destructive testing approaches.

Non-destructive testing techniques (NDTT) involve inspection, testing, or evaluation of materials, assemblies, or components for discontinuities or changes in attributes without altering their quality [10]. Therefore, NDTT techniques do not destroy the sample being characterized, unlike destructive testing. However, in many circumstances, detecting a flaw or characterization of any sample necessitates the application of a combination of NDTTs [14]. In order to safeguard the effectiveness of the evaluation, it is essential to have a good understanding of the background, benefits, and limitations of each NDTT [15]. The NDTT commonly used to characterize nanocarriers are X-ray diffraction, ultraviolet-visible spectroscopy, scanning electron microscopy, dynamic light scattering, atomic force microscopy, confocal laser scanning microscopy, and nuclear magnetic resonance spectroscopy. Table 1 represents different non-destructive techniques used to characterize nanocarriers and their attributes.

**Table 1.** Different non-destructive techniques used to characterize nanocarriers.

| Technique | Nanocarriers Examined | Parameters Examined | References |
|---|---|---|---|
| Scanning electron microscopy | Mesoporous silica materials | Structure/shape, size distribution, and particle size | [16] |
| Scanning electron microscopy | Core-shell nanoparticles of magnetic $Fe_3O_4$-poly (N-isopropylacrylamide) grafted with chitosan | Structure/shape, size distribution, and particle size | [17] |
| Scanning electron microscopy | Ascorbic acid-modified chitosan based superparamagnetic iron oxide nanoparticles | Identification of elemental composition, structure/shape, size distribution, and particle size | [18] |
| Dynamic light scattering | Gelatin nanoparticles | Size distribution, particle size, and surface charge | [19] |
| Dynamic light scattering | Liposomes consisting of phosphatidylcholine or dimyristoylphosphatidylcholine | Size distribution and particle size | [20] |
| Dynamic light scattering | 5-Fluorouracil and Carmofur loaded polyethylene glycol/rosin ester nanocarriers | Size distribution and particle size | [21] |
| X-ray diffraction | Graphene oxide/$Fe_3O_4$ nanocomposite | Phase and structure | [22] |
| X-ray diffraction | Pluronic F127 coated magnetic silica nanocarriers | Crystalline phase and structure | [23] |
| X-ray diffraction | Camptothecin-loaded holmium ferrite nanocarrier | Phase and structure | [24] |
| Atomic force microscopy | Silica nanoparticle | Aggregation topography, Size, shape, and structure | [25] |
| Atomic force microscopy | Chitosan-gum arabic embedded alizarin nanocarriers | Surface topography and uniformity | [26] |
| Atomic force microscopy | Glucosamine-conjugated graphene quantum dots | Surface topography and morphology | [27] |
| Confocal laser scanning microscopy | Polystyrene nanoparticles | Distribution, size, and shape | [28] |

**Table 1.** *Cont.*

| Technique | Nanocarriers Examined | Parameters Examined | References |
|---|---|---|---|
| Confocal laser scanning microscopy | Liposomes | Effect of penetration ability and distribution | [29] |
| Confocal laser scanning microscopy | Miconazole based on chitosan-coated iron oxide nanoparticles | Structure/shape | [30] |
| Nuclear magnetic resonance spectroscopy | Starch nanoparticles | Purity, structure, and composition | [31] |
| Nuclear magnetic resonance spectroscopy | Dehydropeptide nanocarrier | Molecular conformation | [32] |
| Nuclear magnetic resonance spectroscopy | Poly (glycidyl methacrylate)-based double-shell magnetic nanocarrier | Structure and composition | [33] |

As previously mentioned, nanotechnology is a fast-increasing field of study, and many published works on nanocarrier characterization exist. However, just a few mention the NDTT. Most researchers have concentrated on their fundamentals instead of emphasizing the application of characterization approaches. This mini review summarizes the applications of different NDTT used to characterize nanocarriers and their working principles. Readers may gain helpful insights for their studies because of each technique examined. The authors also intend to include possibilities for future development and trends.

## 2. Non-Destructive Testing Techniques

The following section presents the compositional and microstructural characterization of nanocarriers through NDTT along with their working principles, such as X-ray diffraction, dynamic light scattering, confocal laser scanning microscopy, ultraviolet-visible spectroscopy, scanning electron microscopy, atomic force microscopy, and nuclear magnetic resonance spectroscopy.

### 2.1. X-ray Diffraction
2.1.1. Working Principle

X-ray diffraction (XRD) is an analytical technique that discloses important details about a crystalline substance's lattice structure, such as crystallographic structure, chemical composition, bond angles, and unit cell diameters in natural and synthetic materials [34]. The principle of constructive interference of X-rays is used in XRD, which requires a crystalline sample [34]. A beam of X-rays is passed through the specimen and is scattered or diffracted by the atoms in the X-rays' path. Using an appropriately positioned detector, the interference generated by X-ray scattering is seen, and the crystalline structure properties of the material are determined using Bragg's law [7]. According to Bragg's law, the scattering angle is inversely proportional to the interplanar distance at a given wavelength [35].

Using Bragg's Law, the characteristics of dispersed X-rays reveal the crystalline material's arrangement:

$$2d \sin\theta = n\lambda, \tag{1}$$

where n is an integer, $\theta$ is the scattering angle, d is the interplanar, and $\lambda$ is the wavelength.

For crystalline and amorphous materials, sharp and broad diffraction peaks can be seen, with smaller crystallites forming broader peaks [35]. The information obtained from an XRD pattern, as compared to the information obtained from other characterization techniques, allows researchers to determine whether their nanocarriers are uniformly sized or not. To confirm the identity of any analyte, the generation of a specific XRD pattern with characteristic peaks can be compared with XRD patterns from existing literature [36].

### 2.1.2. Application

Sabbagh et al. [37] published a work that used XRD (along with other techniques) to evaluate nanocarriers loaded with metronidazole (MET) made from chitosan (CS) and alginate (AlgNPs). The authors were able to gather a large amount of morphological data by overlaying the patterns of the XRD for CS-AlgNPs, pure MET, and MET-CSAlgNPs. The XRD pattern for pure MET revealed peaks at 2θ = 11.0° and 22.3°. The unloaded CSAlgNPs XRD pattern showed two large peaks at 2θ = 14.9° and 21.6°, suggesting the nanocarriers were amorphous. When these two patterns were compared to the MET-CS-AlgNPs pattern, it is clear that the peaks shown in the pure MET pattern were not present in the MET-CS-AlgNPs pattern. Sabbagh et al. [37] determined that MET was effectively laden in the amorphous portion of the nanocarriers.

Similarly, in another study, XRD was used by Rachmawati et al. [38] to characterize polylactic acid-based nanocarriers loaded with curcumin. The researchers overlaid the XRD patterns obtained for unloaded polylactic acid nanocarriers, pure curcumin, and polylactic acid nanocarriers loaded with curcumin. Pure curcumin's XRD pattern showed sharp, high-intensity peaks, representing significant crystallinity. The peak intensities of nanocarriers were dramatically reduced, indicating lower crystallinity and a probable transfer into an amorphous phase of curcumin. From the results, the authors could determine that curcumin was successfully loaded into their produced polylactic acid nanocarriers.

### 2.2. Ultraviolet-Visible Spectroscopy

### 2.2.1. Working Principle

An ultraviolet-visible (UV-Vis) spectrophotometer is used to determine the absorption or transmission of light in opaque or transparent liquid substances [39]. It compares the intensity of light reflected by a sample to the intensity of light reflected by a reference material. This is accomplished by passing a light beam through the particles dispersed in the liquid medium and then measuring the amount of light that remains in a detector [39].

Nanocarriers are characterized by evaluating their unique absorption peaks caused by a phenomenon known as surface plasma resonance. Each constituent used to create a given nanocarrier will absorb a precise amount of light, like any other matter [40]. A specific spectrum can be formed by utilizing this procedure for a particular fabrication, which may vary depending on the chemicals and contaminants found in the analyte [40]. As a result, the nanocarriers' purity or characteristics can be determined by comparing the obtained spectrum to reference spectrums in other literature.

### 2.2.2. Application

Pang et al. [41] used a UV-Vis spectrophotometer to characterize starch-maleate nanocarriers loaded with curcumin (CurSM). They obtained spectra for each component used in their studies, such as CurSM, free curcumin, and starch malate. The author states that the peaks confirmed the presence of starch-maleate and free curcumin at 250 nm and 420 nm. Furthermore, because distinct absorption peaks for free curcumin were found to rely on the utilized solvent, the authors concluded that free curcumin attained tautomeric equilibrium with keto-enol and diketo forms.

Similarly, by observing a distinctive peak at 520 nm, Hung et al. [42] proved the usefulness of UV-Vis spectroscopy in demonstrating the favorable loading of an anticancer drug into gold nanocarriers based on collagen. The UV-Vis technique was also beneficial for the characterization of gelatin-stabilized copper nanocarriers, according to Musa et al. [43]. The authors reported that the intensity of the peak at 583 nm grew steadily when the concentration of gelatin used was increased. When the spectral profiles of gelatin-stabilized copper nanocarriers with different gelatin concentrations were superimposed, a noticeable red shift from 583 nm to 590 nm was observed, implying that the size of the copper nanocarriers increased with gelatin concentration.

### 2.3. Scanning Electron Microscopy with Energy Dispersive X-ray Spectroscopy

2.3.1. Working Principle

A scanning electron microscope (SEM) images a sample by scanning it in a raster scan pattern with a high-energy electron beam [34]. The interaction of those electrons with the atoms in the sample produces signals that carry information about the sample's composition, surface topography, and other qualities, such as electrical conductivity [44]. SEM photographs of a sample surface can be very high resolution, revealing features as small as 1 to 5 nm [39]. Electrons created by a source are accelerated in a field gradient under a vacuum. The beam is focused onto the specimen via electromagnetic lenses. The specimen emits several signals because of its reaction with the electron beam. A detector catches the secondary electrons, and the intensity of these secondary electrons is compared to the scan of primary electron beam to create an image of the sample surface. Finally, the image is displayed on a computer monitor [39]. Figure 2 shows the major components of an SEM.

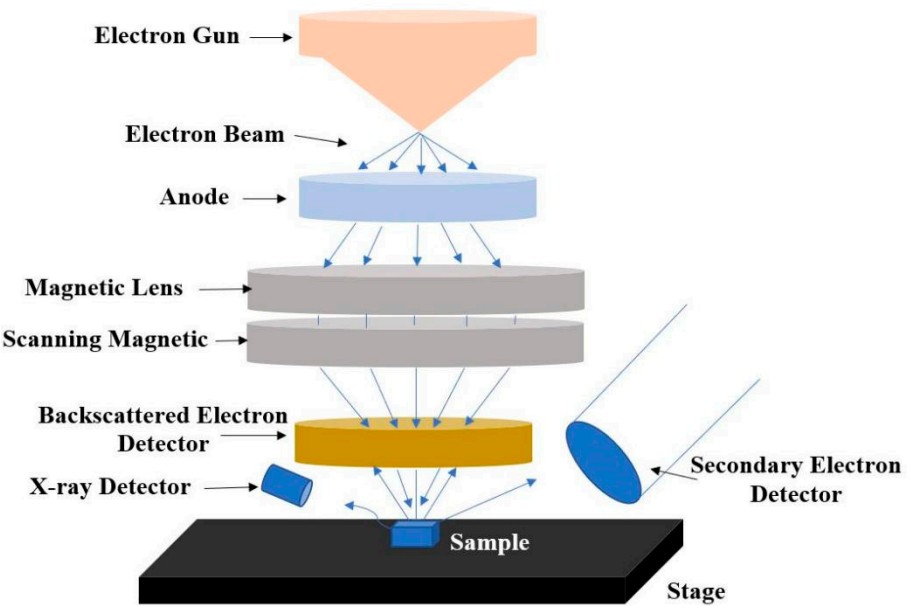

**Figure 2.** Schematic representation of a scanning electron microscope.

The elemental analysis or chemical characterization of a material is performed using energy-dispersive X-ray spectroscopy (EDS or EDX). It is a type of X-ray fluorescence spectroscopy that investigates a sample through interactions between electromagnetic radiation and matter, examining X-rays generated by the matter as a result of hitting the charged particles [34]. It provides a wide range of characterization capabilities because of the fundamental premise that each element has a unique atomic structure, which allows X-rays indicative of an element's atomic structure to be separated from others. Researchers can learn a lot about the elemental composition of nanocarriers from the combined results of a single SEM-EDX investigation [45].

2.3.2. Application

Rezaei et al. [46] used SEM to investigate the structural properties of nanocarriers derived from hyaluronic acid-functionalized with lipoic acid and chitosan (HACSLA-NPs). These nanocarriers had a diameter ranging from 50 to 130 nm and an orderly distribution and arrangement. The identity of HACSLA-NPs was revealed by further EDX microanalysis, which revealed distinct peaks in the EDX spectra, indicating the existence of NO and hydrogen components. Lipoic acid is made up of Sulphur (S), which was confirmed by peaks and the favorable loading of lipoic acid. The authors were able to describe nanoparticles using SEM-EDX without inflicting any damage to the materials.

Similarly, Musa et al. [43] reported SEM photos of gelatin-based copper nanoparticles. According to the author's SEM image analysis, the gelatin matrix securely sustained the copper nanoparticles. These results demonstrated that as the concentration of the gelatin was increased, the mean diameters of the nanoparticles gradually dropped, resulting in the particles becoming smaller. This is an anticipated result based on the findings obtained from the various characterizations carried out on the copper nanoparticles.

### 2.4. Nuclear Magnetic Resonance Spectroscopy

2.4.1. Working Principle

Nuclear magnetic resonance (NMR) spectroscopy works on the principle that many atomic nuclei comprise spin and are charged electrically [47,48]. An energy transfer from the base energy to a higher energy level is achievable if an external magnetic field is provided (generally a single energy gap). When the spin returns to its base level, energy is emitted at the same frequency. The signal corresponding to this transfer is measured and processed to produce an NMR spectrum for the nucleus [48].

2.4.2. Application

Alp et al. [31] assessed nanocarriers based on starch and loaded with histone deacetylase inhibitors (CG-1521). They demonstrated the recent use of NMR spectroscopy to describe nanocarriers. Similarly, Anantachaisilp et al. [49] used 1H-NMR spectroscopy to characterize lipid-based nanocarrier loaded with γ-oryzanol in drug–lipid interaction research. Their research describes the structural and chemical examination of γ-oryzanol loading within a model lipid nanoparticle drug delivery system made up of Cetyl palmitate (solid lipid) and Miglyol 812 (liquid lipid). The 1H-NMR study revealed that the chemical shifts of the liquid lipid in γ-oryzanol loaded systems were at a higher field than those in γ-oryzanol free systems, indicating incorporation of γ-oryzanol in the liquid lipid. The authors were able to demonstrate the effective incorporation of Miglyol 812 onto the lipid based nanocarrier by analyzing the NMR spectra of their nanocarriers and the existence of peaks that indicate the presence of medium chain triglycerides.

### 2.5. Dynamic Light Scattering

2.5.1. Working Principle

Dynamic Light Scattering (DLS) is a commonly used method for determining particle size. It is considered a hydrodynamic method because it directly evaluates quantitative hydrodynamic values of translational and rotational diffusion coefficients, which correspond to particle sizes and shapes via theoretic relationships [50]. It analyses the modulation of the intensity of scattered light as a function of time to determine the hydrodynamic size of particles using the mechanism of light scattering from a laser passing through a colloidal solution [51].

The Brownian movements of the particles in the solvent behave unpredictably, as specified by a translational diffusion coefficient [52]. Particle size also affects this coefficient; the smaller the particles, the faster they move. High temperatures accelerate Brownian motion. As a result, DLS examines the velocity distribution of particles moving due to Brownian motion as dynamic variations in light scattering intensity [52]. The particle sizes are then measured or characterized using Stokes–Einstein equations, in which the translational self-diffusion coefficient of spherical particles is proportional to the particle radius [53]. Table 2 presents the advantages and disadvantages of different non-destructive characterization techniques.

**Table 2.** Advantages and disadvantages of different non-destructive characterization techniques.

| Techniques | Advantages | Disadvantages | References |
|---|---|---|---|
| Scanning Electron Microscopy (SEM) | • The image produced can have a resolution of up to 100,000×.<br>• Nanomaterials' size/size distribution and shape can be measured directly. | • Sample preparation can be time consuming.<br>• High cost of equipment. | [54–56] |
| X-ray diffraction | • The average crystallite size can be determined.<br>• Well-understood technology with exceptional spatial resolution. | • Only single conformation/binding state of sample is accessible.<br>• Limited applications in crystalline materials. | [57–59] |
| Nuclear magnetic resonance spectroscopy (NMR) | • Little sample preparation required. | • NMR is limited to examining a few nuclei.<br>• The procedure is time consuming. | [60,61] |
| Dynamic Light Scattering | • Easy sample preparation.<br>• Suitable for monodisperse samples. | • Complicated interpretation of data.<br>• To avoid multiple scattering, sample preparation requires proper dilution as well as pH and viscosity adjustments. | [50,62] |
| Confocal laser scanning microscopy | • Ability to generate optical pictures at various depths in a noninvasive way, without the need for mechanical sample sectioning.<br>• It is a very useful technique for capturing high-resolution images of biological and biomaterial specimens. | • Any unentrapped dye in the samples should be removed; otherwise, the outcome may be influenced.<br>• Requires use of a fluorescent dye to stain the samples that may interfere with results. | [28,29,63] |
| Atomic force microscopy | • It generates 3-D and topographical photographs of the samples in high quality.<br>• It is used to characterize and visualize the in vivo delivery and release of genetic materials. | • The contact and tapping modes of analysis can result in sample damage and contamination of the tip.<br>• Expensive and time-consuming sample analysis and preparation. | [62,64–67] |

### 2.5.2. Application

Boufi et al. [68] used a combination of atomic force microscopy, DLS, and SEM techniques to assess the morphological properties of starch-based nanocarriers. After subjecting the manufactured starch nanocarriers to sonication (for 75 min), the authors reported that Starch granules were not visible in SEM images; instead, enormous aggregates of nanoparticles ranging in size from 30 to 40 nm were discovered. The authors hypothesized that the high number of -OH groups on the nanocarriers' surface, the possibility of water evaporation, and the nanocarriers' small size were responsible for their self-aggregating tendency. The nanoparticle size distribution (30–40 nm) was consistent with the data acquired from the DLS analysis.

Similarly, Meyabadi et al. [69] used an enzymatic hydrolysis technique to create cellulose nanoparticles from cotton waste. DLS determined that the average particle size of the hydrolyzed cotton nanoparticles was less than 100 nm. Likewise, Huang et al. [70] produced cross-linked nanoparticles of gene delivery made up of triphosphate and polyethylene amine. The size distribution and average diameter of nanoparticles was investigated using

the DLS. The results showed that nanoparticles had a uniform particle size with an average particle size of 120 nm.

*2.6. Confocal Laser Scanning Microscopy*

2.6.1. Working Principle

A confocal laser scanning microscope (CLSM) takes an image of a slice of a sample while avoiding out-of-focus adjacent planes [71]. The ability to focus on a specific focal point, known as depth discrimination or optical sectioning, distinguishes it from a traditional microscope. Image production with a CLSM consists of two lens systems: the collector lens and the objective lens, which sees the picture plane. The collector lens is used as an image lens rather than a collecting lens, using a point detector with a very small area [71]. The imaging and detecting systems are on the same focal plane as a whole, which is why this setup is called "confocal." This microscope's advanced version is perfect for converting the specimen into a three-dimensional representation using volume visualization methods [72].

2.6.2. Application

There have been numerous reports of CLSM's possible application in nanoparticle-based drug delivery. For example, CLSM was used by Alvarez-Román et al. [28] to investigate the penetration and distribution of FITC-loaded polymeric nanoparticles in the epidermal layer. The authors were able to visualize the dispersion of non-biodegradable, fluorescent polystyrene nanoparticles (diameters 20 and 200 nm) over porcine skin with the use of the equipment.

Similarly, CLSM was used by Verma et al. [29] to investigate the deposition of carboxyfluorescein dye across multiple layers of human abdomen skin from a series of liposomes with particle sizes ranging from 120 to 810 nm. The researchers discovered that the size of the liposomes had an inverse relationship with their skin deposition. Likewise, Zhang et al. [73] used CLSM to evaluate the potential of ethosomes to transport 5-fluorouracil to a human hypertrophic scar. Ethosomes are nanoparticles formed from phospholipids and ethanol at the proper concentration.

Hence, CLSM has been employed in multiple studies on cell absorption, nanoparticle transdermal penetration mechanism, and skin structure elucidation after treatment with the formulation [74,75].

*2.7. Atomic Force Microscopy*

2.7.1. Working Principle

Atomic force microscopy (AFM) may be utilized to test both non-conducting and conducting substances. It produces a three-dimensional and topographical image of the substance being examined [64]. The calculation of ultra-small forces on the sample particles, which can be as tiny as single atoms, is used to generate a picture using AFM. A metal tip connected to a cantilever scans the specimen's surface at a static tunnelling current, with the displacement of the tip controlled by the voltage given to the piezo drives. As the tip approaches the surface, electrostatic and magnetic forces and interatomic interactions between the sample surfaces held on the tip and the substrate causes the cantilever to deflect according to Hooks law. The cantilever's deflection in relation to the sample's elevated or lowered features is then used to generate a topographic image of the sample [64].

2.7.2. Application

Theodoropoulos et al. [20] used AFM to examine the forms and sizes of liposome molecules containing carborane-loaded DMPC (1, 2-dimyristoyl-sn-glycero3-phosphocholine), carborane-loaded PC, carborane-free DMPC, and carborane-free phosphatidylcholine (PC). Similarly, Müller et al. [65] have discussed the many uses of AFM in nanobiotechnology with their study's bioanalyte identification in the medical diagnostics, picomolar range, and environmental monitoring.

Wan et al. [76] used AFM to investigate the dynamics of DNA release from a reducible polyplex consisting of nuclear localization signal (NLS[6+]) peptide. The AFM investigation shows that DNA can be released from reducible polyplexes when they are subjected to conditions that replicate physiological reduction processes. The reducible polyplexes offer an excellent platform on which to conduct real-time AFM research on the kinetics of DNA decondensation. The polyplexes are able to maintain their stability in the oxidizing environment that is characteristic of the nonreducing extracellular space.

According to Sarwar et al. [77], AFM is important in analyzing the texture and structure of chitosan films, which include a synthetic polymer called polyallylamine hydrochloride (PAH). After analyzing the AFM micrographs, they recognized the uneven texture of films with increasing concentrations of chitosan. Likewise, Weiss et al. [19] showed how AFM imaging could characterize gelatin-based nanocarriers. The researchers used a mix of SEM and AFM to determine the overall morphological impression. Nanocarriers based on gelatin in deionized water were studied using AFM height micrographs. The authors could recognize the near-spherical form of the produced nanocarriers after further study. Figure 3 shows a schematic representation of AFM.

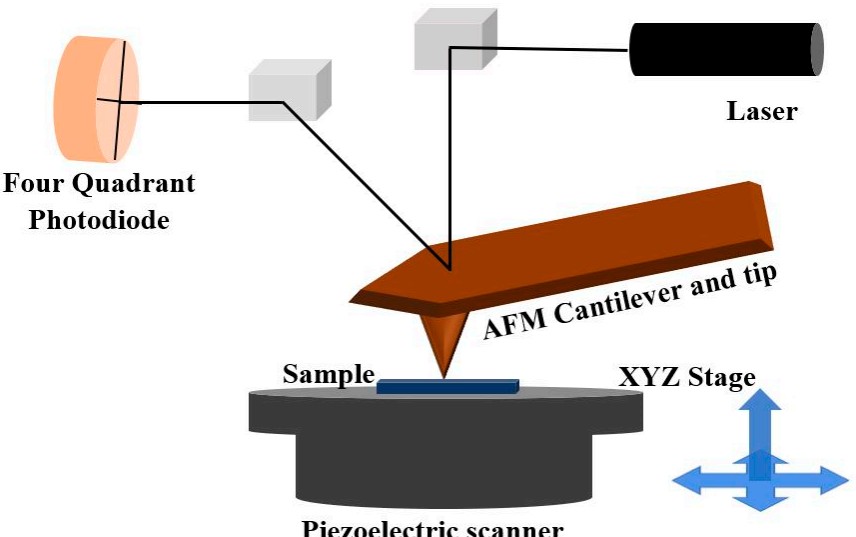

**Figure 3.** Schematic representation of an atomic force microscope.

### 3. Future Perspectives

Over other destructive testing procedures, NDTT has several advantages. Because the tests are reproducible and a multitude of tests can be used in conjunction to correlate findings, the methods in this category are highly accurate inspection procedures. However, a mix of approaches will be needed to characterize nanocarriers effectively.

Furthermore, by developing more advanced, effective, and precise techniques, such as machine vision techniques and artificial intelligence, the future of using NDTT for nanocarrier characterization will improve the evaluation of internal quality parameters. Presently, the initial costs of such instruments are high. Hence, they can be purchased with collaboration or allocation of research grants by multiple research institutes to act as the main focus of investigation for investigators working in a specific region.

Characterization of nanocarriers confirms the physicochemical parameters related to the physiological performance but does not account for their toxicity. The safety of nanomaterials and their risks have yet to be investigated, necessitating further risk assessment. Aside from high efficiency and targeted delivery via nanocarriers, more research is needed to develop pharmacological formulations to treat life-threatening diseases. The direct and indirect effects of nanomaterials on human health must be investigated. In addition, it is critical to establish regulatory controls to protect the public from the potential adverse effects of nanotechnology [78]. As a result, it is highly recommended that NDTT be

developed to assess the toxicity of nanocarriers in the future. Furthermore, it is evident that the current characterization techniques are insufficient, as they have numerous flaws. We hope that many newer techniques will emerge in the coming days to replace the existing advanced characterization techniques and answer many of the questions that researchers face today.

## 4. Conclusions

Nanomaterials suited for carrying pharmaceuticals have been rapidly established in current years to improve drug therapy and reduce adverse effects. Hence, nanocarrier characterization is crucial for controlling their anticipated in vivo and in vitro action, ensuring their safe application. Several non-destructive approaches such as AFM, SEM, DLS, and others are available to characterize various nanocarriers. Each NDTT described in this manuscript has its own set of comparative advantages and disadvantages for the characterization of various nanocarrier systems. Researchers must choose the appropriate procedures for the characterization of a potential nanocarrier based on their objectives. In addition, combining these techniques in the right way can offer the data needed to understand their pharmacokinetics and drug-release characteristics with new ideas for improving drug delivery systems. However, even though advanced analytical and visualization tools have examined many elements of nanoscience and nanotechnology, basic difficulties and loopholes, such as an error in sample preparation, duration of sample preparation, improper handling of the instruments, wrong data interpretation, etc., still exist. To adequately address these issues, an open-minded and multidisciplinary approach and thorough knowledge of each instrument are required. We hope that many newer techniques will emerge in the coming days to replace the existing characterization techniques and solve many problems today's researchers face.

**Author Contributions:** Conceptualization, R.I.B., H.K.K. and S.R.; methodology, R.I.B. and H.K.K.; investigation, R.I.B. and H.K.K.; resources, A.E., W.R. and A.S.; data curation, R.I.B.; writing—original draft preparation, R.I.B., H.K.K. and S.R.; writing—review and editing, M.A., A.E., W.R. and A.S.; supervision, A.E., W.R., M.A. and A.S.; project administration, A.S.; funding acquisition, A.E. and A.S. All authors have read and agreed to the published version of the manuscript.

**Funding:** This research was funded by Natural Sciences and Engineering Research Council of Canada (NSERC) (Grant # RGPIN-2017-03975).

**Institutional Review Board Statement:** Not applicable.

**Informed Consent Statement:** Not applicable.

**Data Availability Statement:** Not applicable.

**Acknowledgments:** The authors would like to express their gratitude to the Natural Sciences and Engineering Research Council of Canada (NSERC) for funding this study (Grant # RGPIN-2017-03975).

**Conflicts of Interest:** The authors declare no conflict of interest.

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
