# Peer review of "Application of Non-Destructive Testing Techniques (NDTT) to Characterize Nanocarriers Used for Drug Delivery: A Mini Review"

_biophysica, doi:10.3390/biophysica2030016_

Round 1

Reviewer 1 Report

The manuscript, generally, lacks significance. Sentences lack cohesion. Too often, meaning is ambiguous. Should be less vague, more precise. It should state aim more clearly. It should stress on utility of the review.

The authors admit comprehensive reviews already exist. I don't think it provides additional scientific value.

It overflows with rather general knowledge. There is no particular insight or innovative proposition.

68% of the references are more than 5 years old.

24 out of 61 references are of other reviews or book chapters describing methodology.

Author Response

Comment 1: The manuscript, generally, lacks significance. Sentences lack cohesion. Too often, meaning is ambiguous. Should be less vague, more precise. It should state aim more clearly. It should stress on utility of the review. The authors admit comprehensive reviews already exist. I don't think it provides additional scientific value. It overflows with rather general knowledge. There is no particular insight or innovative proposition.

Response:  We thank the reviewer for the valuable comments. According to the suggestion, the manuscript has been revised to be more precise and less vague.

Comment 2: 68% of the references are more than 5 years old.

Response:  We thank the reviewer for the valuable comments. Since it was a comprehensive review, research articles from the previous ten years were considered. Significant modification has been made to the manuscript to make it current and up-to-date.  

Comment 3: 24 out of 61 references are of other reviews or book chapters describing methodology

Response:  The purpose of reviewing reviews and book chapters was to collect general information, such as the definitions, working principles, and advantages and disadvantages of different NDTT methodologies.

Reviewer 2 Report

Dear Authors,

The manuscript entitled "Application of non-destructive testing techniques (NDTT) to characterize nanocarriers used for drug delivery: A mini review" provides the readers with an overview of the techniques that are useful for characterizing the nanocarriers. The fundamentals of the techniques, their applications, and some examples are given in this review. 

The manuscript is well-written and well-organized, and relevant references have been cited. The introduction provides the background about nanocarriers and their characterization before going deeper into the non-destructive techniques for characterizing the nanocarriers. 

I just have one comment:

- Please review line 19 in the abstract. There are some non-connected words: 

In recent years, fields of, nanomaterials. This mini review provides an overview on applications of [...].

Kind regards,

Author Response

Comment: The manuscript entitled "Application of non-destructive testing techniques (NDTT) to characterize nanocarriers used for drug delivery: A mini review" provides the readers with an overview of the techniques that are useful for characterizing the nanocarriers. The fundamentals of the techniques, their applications, and some examples are given in this review. 

The manuscript is well-written and well-organized, and relevant references have been cited. The introduction provides the background about nanocarriers and their characterization before going deeper into the non-destructive techniques for characterizing the nanocarriers. 

I just have one comment:

- Please review line 19 in the abstract. There are some non-connected words: 

In recent years, fields of, nanomaterials. This mini review provides an overview on applications of [...].

Response:  We thank the reviewer for the valuable comments. As per the suggestions, all the changes have been made.

Reviewer 3 Report

The authors have done a satisfactory work, and i suggest its publication after working out the major comments.

1. The authors should provide the abbreviations wherever they used in the manuscript, like NDTT, they used in title but never defined in the abstract or introduction.

2.In keywords, please include NDTT as the authors used multiple times throughout the manuscript.

3. The authors are requested to work on the tables and include more and latest references and make it properly aligned.

4.The Introduction part lacks proper citations.

5.There are multiple grammatical mistakes that can easily be rectified after proof read.

Author Response

The authors have done a satisfactory work, and I suggest its publication after working out the major comments.

Comment 1: The authors should provide the abbreviations wherever they used in the manuscript, like NDTT, which they used in the title but never defined in the abstract or introduction.

Response: We thank the reviewer for the valuable comments. As per the suggestions proper abbreviations have been included in the manuscript.

Comment 2: In keywords, please include NDTT as the authors used multiple times throughout the manuscript.

Response: As per the suggestion, changes have been made in the manuscript.

Comment 3: The authors are requested to work on the tables and include more and latest references and make it properly aligned.

Response: As per the suggestion latest references have been included with proper alignment.

Comment 4: The Introduction part lacks proper citations.

Response: As per the suggestion citation have been included.

Comment 5: There are multiple grammatical mistakes that can easily be rectified after proofreading.

Response: As per the suggestion, changes have been made. Authors have tried their best to improve the overall readability and clarity of the manuscript.

Round 2

Reviewer 1 Report

I appreciate that authors have operated some changes in order to improve the manuscript, but I think the impact of these changes is minor. I don't think my objections were handled. I understand that meeting my objections would have meant rewriting the manuscript, yet I still feel against its publishing.

The main flaws refer to lack of novelty, a vague approach on the subject and a resemblance to a collection of rather basic general knowledge.

Reviewer 3 Report

The work can now be accepted